# Endobronchial Ultrasound-Guided Transbronchial Forceps Biopsy: A Retrospective Bicentric Study Using the Olympus 1.5 mm Mini-Forceps

**DOI:** 10.3390/jcm11164700

**Published:** 2022-08-11

**Authors:** Fabienne Rüber, Gilles Wiederkehr, Carolin Steinack, Sylvia Höller, Peter Karl Bode, Fabian Kölbener, Daniel Peter Franzen

**Affiliations:** 1Department of Pulmonology, University Hospital Zurich, Raemistrasse 100, 8091 Zurich, Switzerland; 2Respiratory Unit, Hirslanden Clinic St. Anna, St. Anna-Strasse 32, 6006 Lucerne, Switzerland; 3Institute of Clinical Pathology, Stadtspital Zurich, Birmensdorferstrasse 497, 8063 Zurich, Switzerland; 4Department of Pathology, University Hospital Zurich, Raemistrasse 100, 8091 Zurich, Switzerland; 5Department of Pathology, Cantonal Hospital Winterthur, Brauerstrasse 15, 8401 Winterthur, Switzerland; 6Department of Internal Medicine, Spital Burgdorf, Oberburgstrasse 54, 3400 Burgdorf, Switzerland; 7Department of Internal Medicine, Spital Uster, Brunnenstrasse 42, 8610 Uster, Switzerland

**Keywords:** transbronchial forceps biopsy, transbronchial needle aspiration, endobronchial ultrasound, diagnostic yield

## Abstract

When evaluating mediastinal/hilar lymphadenopathy (LAD) or masses, guidelines recommend endobronchial ultrasound (EBUS)-guided transbronchial needle aspiration (TBNA) as an initial technique for tissue analysis and diagnosis. However, owing to the small sample size obtained by needle aspiration, its diagnostic yield (DY) is limited. EBUS transbronchial forceps biopsy (TBFB) used as a complimentary technique to EBUS-TBNA might allow for better histopathological evaluation, thus improving DY. In this retrospective bicentric study, we assessed the DY and safety of an EBUS-guided 1.5 mm mini-forceps biopsy combined with EBUS-TBNA for the diagnosis of mediastinal/hilar LAD or masses compared to EBUS-TBNA alone. In total, 105 patients were enrolled. The overall DY was 61.9% and 85.7% for TBNA alone and EBUS-TBNA combined with EBUS-TBFB, respectively (*p* < 0.001). While the combined approach was associated with a significantly higher DY for lung cancer diagnosis (97.1% vs. 76.5%, *p* = 0.016) and sarcoidosis (85.2% vs. 44.4%, *p* = 0.001), no significant differences in DY were calculated for subgroups with smaller sample sizes such as lymphoma. No major adverse events were observed. Using a 1.5 mm mini-forceps is a safe and feasible technique for biopsy of mediastinal or hilar LAD or masses with superior overall DY compared to EBUS-TBNA as a standalone technique.

## 1. Introduction

Endobronchial ultrasound (EBUS)-guided transbronchial needle aspiration (TBNA) is established as a safe and minimally invasive technique for cytological examination of mediastinal and hilar lymphadenopathy (LAD) with a high diagnostic yield (DY) [1,2,3]. According to several meta-analyses, the reported sensitivity of EBUS-TBNA is 88–93% for the staging [4,5] and 92% [6] for the diagnosis of non-small-cell cancer (NSCLC). Therefore, EBUS-TBNA is recommended as the lung cancer staging modality of choice over mediastinoscopy [7,8,9]. However, there are limitations of this technique, mainly due to the small amount of tissue sampled by needle aspiration which allows only for cytological analyses in most of the cases. In particular, molecular testing for treatable oncogenic driver mutations or testing for programmed cell death ligand-1 (PD-L1) expression requires larger amounts of sample volume. In addition, the DY of EBUS-TBNA for the diagnosis of malignant lymphoma and sarcoidosis remains inferior compared to surgical biopsy, with a reported sensitivity of 79% [10] for sarcoidosis and approximately 66% [11,12,13] for lymphoma, with some studies showing yields as up to 91% [14,15]. Thus, the gold standard for diagnosis of lymphoma remains a surgical biopsy by mediastinoscopy or thoracoscopy. However, a less invasive approach performed in an outpatient setting would be highly desirable from the patient’s or payer’s perspective.

To overcome the abovementioned limitations, the use of an EBUS-guided transbronchial forceps biopsy (TBFB) has been proposed and evaluated in several studies yielding improved diagnostic accuracy [1,12,16,17,18]. A recent meta-analysis reported a statistically significant increased DY for lymphoma (86% vs. 30%) and for sarcoidosis (93% vs. 58%) in patients with intrathoracic LAD who underwent additional tissue acquisition using EBUS-TBFB following EBUS-TBNA [18]. However, despite these promising results, EBUS-TBFB has not been implemented into clinical routine. With the market launch of the 1.5 mm mini-forceps by Olympus (Olympus Medical Systems Corp., Tokyo, Japan), which was originally manufactured for radial probe EBUS-guided biopsy of peripheral lung lesions, there might be a new instrument and technique for tissue acquisition from mediastinal/hilar LAD or masses. There are only limited data on the use of a 1.5 mm mini-forceps for EBUS-guided TBFB of mediastinal/hilar LAD or masses. In 2012, Herth et al. demonstrated that using a 1.5 mm mini-forceps to obtain tissue for diagnosis of enlarged mediastinal LAD is a safe and feasible technique, which provided a diagnosis in 86% of cases [19]. Recently, Mehta et al. showed promising results in a small cohort of 30 patients using the 1.5 mm mini-forceps from Olympus [20]. The aim of the present study was to investigate the DY, safety, and feasibility of EBUS-TBFB using the 1.5 mm mini-forceps in addition to EBUS-TBNA as compared to standard EBUS-TBNA as a standalone technique for diagnosis of mediastinal/hilar LAD or masses in a bicentric study with a large cohort.

## 2. Materials and Methods

### 2.1. Patient Selection and Overall Study Design

For the present retrospective study, we enrolled all patients with mediastinal/hilar LAD or masses detected by chest computed tomography (CT) who underwent EBUS-TBNA and subsequent EBUS-TBFB performed by three experienced interventional pulmonologists (G.W., C.S., and D.P.F) at the University Hospital Zurich or at the Clinic St. Anna, Lucerne from 1 January 2020 to 31 December 2020 when sarcoidosis, lymphoma, or infection including tuberculosis was suspected, or when sufficient tissue was required for cancer staging or diagnosis. Patients were included if they were 18 years or older and written informed consent was obtained.

The primary outcome was the overall DY of EBUS-TBFB in addition to EBUS-TBNA compared to EBUS-TBNA alone. The secondary outcome was safety (bleeding, mediastinal infection, and pneumothorax) and technical feasibility. The study was approved by the Competent Ethics Committee of the Cantons of Zurich (BASEC-ID 2019-02479) and Lucerne (BASEC-ID 2020-00136-L).

### 2.2. Procedure and Sedation

All bronchoscopies were performed under either moderate sedation or general anesthesia according to the examiner’s decision in in- or outpatient setting. Following airway inspection using a flexible Olympus bronchoscope (190 series), the lymph node stations 2, 4, 7, 10, and 11 from both sides, in addition to a possible mediastinal/hilar mass, were systematically assessed using an Olympus EBUS bronchoscope (BF-UC180F, Olympus Medical Systems Corp., Tokyo, Japan). Lymph nodes with a diameter equal or greater than 10 mm were sampled with three passes with a minimum of 10 advances per site using a 19-gauge needle (NA-U402SX-4019, Olympus Medical Systems Corp., Tokyo, Japan) or a 22-gauge needle (NA-201Sx-4022, Olympus Medical Systems Corp., Tokyo, Japan) according to the discretion of the bronchoscopist.

After identification of the target lymph node or mass by EBUS for subsequent TBFB according to size and appearance of malignancy, another pass with the EBUS needle was added. In order to augment the puncture hole, the needle sheath was advanced through the bronchial wall into the lymph node/lesion and moved forward and backward several times. After extraction of the EBUS needle and sheath, respectively, a closed mini-forceps (FB-433D, Olympus Medical Systems Corp., Tokyo, Japan) was advanced through the working channel of the EBUS bronchoscope to re-enter the lymph node or mass through the pre-existing puncture hole while avoiding any movement of the bronchoscope. After penetration of the capsule of the lymph node, the forceps was opened and advanced until reaching the distal end of the lymph node. At the distal end, the forceps was closed and pulled for each specimen. This procedure was repeated four to five times under real-time imaging using EBUS. Finally, the closed mini-forceps was withdrawn through the working channel.

### 2.3. Cytological and Histological Analyses

Cytological specimens obtained using EBUS-TBNA were processed per institutional protocol and examined on a per node basis. The specimens were preserved in normal saline and immediately transferred to the respective Institutes of Pathology.

EBUS-TBFB specimens were processed as histology specimens and placed into formalin solution for permanent fixation. All specimens were analyzed by pathologists at the pathology departments of the University Hospital Zurich and at the Lucerne Cantonal Hospital, respectively.

### 2.4. Safety

All patients were routinely screened for pneumothorax with immediate lung ultrasound (LUS) and a chest radiograph (CXR) within 2 h post procedure. In addition, patients were asked to measure their body temperature daily during 7 days after the intervention and report temperatures above 38.5 °C lasting for more than 24 h to screen for a possible pneumonia or mediastinitis.

### 2.5. Reference Standard and Endpoints

The primary endpoint was the DY of EBUS-TBNA and EBUS-TBNA combined with EBUS-TBFB overall and among the subgroups. Secondary endpoints were its technical feasibility and safety. The DY was defined as the percentage of patients for whom at least one of the respective samples (either after TBNA alone or TBNA and TBFB combined) was diagnostic. At least one of the samples of TBNA or TBFB had to be diagnostic to define the procedure as diagnostic. The detection of atypical or highly suspicious cells was not deemed diagnostic and, therefore, considered as negative cases. If EBUS-TBNA or EBUS-TBFB samples correctly excluded an alternative diagnosis, e.g., the exclusion of sarcoidosis, the procedure was also considered diagnostic.

If EBUS-TBNA or EBUS-TBFB specimens were not diagnostic or suitable for analysis, the final diagnosis was made either by surgical biopsy (mediastinoscopy or through sampling at another site), by re-bronchoscopy or by follow-up.

### 2.6. Statistical Analysis

Descriptive statistics were used to illustrate demographics and patient characteristics. Groups were compared with McNemar’s test using the SPSS version 26.0.0.0. (SPSS, Chicago, IL, USA). Statistical significance was defined as a *p*-value < 0.05.

## 3. Results

In 1 year, 179 patients were enrolled in the study. A total of 36 patients were excluded because EBUS-TBFB was technically not possible, mostly due to the inability to penetrate the bronchial wall. Among these 36 patients, the 19-gauge needle was used in 28 patients (77.8%). Both TBNA and TBFB were successfully conducted in the remaining 143 patients, of which 105 (64.8% males with a mean age of 63.1 years (±13.6)) were totally included (Figure 1). Most procedures were performed under general anesthesia (53.3%) and on an outpatient basis (55.2%). The mean size of the lymph nodes or tumor biopsied was 18.1 mm (range 10.0 mm–42.0 mm) and 32.8 mm (range 13.0–60.0 mm), respectively. On average, 2.7 (range, 1–6, SD = ±1.1) lymph nodes were sampled per EBUS-TBNA procedure. In 11 cases, a hilar/mediastinal mass was sampled by EBUS-TBNA. On average, 1.01 (range, 1–3; SD = ±0.3) lymph nodes were sampled by TBFB. Analogous to TBNA, in 11 cases, a hilar/mediastinal mass was sampled by TBFB. The distribution and size of lymph nodes and tumors examined by TBNA and TBFB are presented in Table 1. Patient characteristics and demographics are presented in Table 2.

Indications for EBUS-TBNA and EBUS-TBFB, separated by the final diagnoses, are presented in Table 3. The most frequent final diagnosis was lung cancer in 45 patients (42.9%) followed by sarcoidosis in 27 (25.7%), malignant lymphoma in eight (7.6%), malignancy other than lung cancer in seven (6.7%, metastasis of esophageal cancer *n* = 2, thymus carcinoma *n* = 1, melanoma *n* = 1, breast cancer *n* = 1, sarcoma *n* = 1, and ovarian carcinosarcoma *n* = 1), and other diagnosis in 18 (17.1%) patients. Other diagnoses included infectious diseases, inflammatory diseases, pneumoconiosis, and interstitial lung diseases.

### 3.1. Diagnostic Yield

Overall, EBUS-TBNA as a standalone procedure yielded the final diagnosis in 61.9% of the patients compared to 85.7% when combining EBUS-TBNA and EBUS-TBFB, resulting in a significant difference in DY between the two techniques (*p* < 0.001). Thus, the combination of EBUS-TBFB and EBUS-TBNA provided additional or superior diagnostic information in 25 patients (23.8%) as compared to EBUS-TBNA alone (sarcoidosis *n* = 11, lung cancer *n* = 7, other diagnoses *n* = 5, one lymphoma, and one malignancy other than lung cancer). Conversely, in 10 patients (9.5%), the diagnosis was exclusively established by TBNA (lung cancer *n* = 4, lung cancer staging *n* = 2, sarcoidosis *n* = 2, and other diagnoses *n* = 2). Overall, no significant difference in DY between 19-gauge TBNA biopsies and 22-gauge TBNA biopsies was calculated (58.8% for 19-gauge TBNA and 64.8% for 22-gauge TBNA, *p* = 0.53).

Overall, EBUS-TBFB failed to obtain adequate histology specimens in 10 (9.5%) cases. A total of 15 patients with a nondiagnostic bronchoscopy after TBNA and TBFB sampling underwent further diagnostics or follow-up to provide a classifying diagnosis in all patients (Figure 2).

After subgroup analysis of the final diagnoses, a significantly higher DY of EBUS-TBFB combined with EBUS-TBNA was calculated for lung cancer diagnosis (97.1% vs. 76.5%, *p* = 0.016) and sarcoidosis (85.2% vs. 44.4%, *p* = 0.001). Among the other subgroups, the DY between the two procedures was similar (Table 4).

### 3.2. Adverse Events

There was neither a postinterventional pneumothorax nor evidence of mediastinal infection after the procedures. In addition, no major bleeding was observed. However, minor bleeding occurred in 21 patients (20%), of which 18 were treated with topical use of vasoconstrictors. In two patients, there was respiratory failure requiring noninvasive ventilation in one case and hospitalization for observation in the other case.

## 4. Discussion

In this retrospective bicentric study, we tested for the DY and safety of EBUS-TBFB in addition to EBUS-TBNA as a complementary sampling technique using a 1.5 mm forceps for diagnostic tissue acquisition from mediastinal/hilar LAD or masses compared to standard EBUS-TBNA alone. We were able to demonstrate a significantly higher overall DY of the combined technique as compared to EBUS-TBNA alone in a large cohort of 105 patients. In the subgroup analysis, we found a significantly superior DY for the diagnosis of lung cancer and sarcoidosis. However, EBUS-TBFB was technically not feasible in a significant number of patients.

Basically, it is not surprising that histological specimens yield higher diagnostic accuracy as compared to cytology. For this reason, video-assisted mediastinoscopy (VAM) is recommended according to the guidelines by the European Society of Thoracic Surgery (ESTS) when EBUS/EUS-TBNA remains nondiagnostic [21]. However, VAM is an invasive procedure which can be avoided when endoscopic techniques with comparable DYs are available. There are substantial data on the diagnostic accuracy of EBUS-TBNA.

In a recent meta-analysis by Agrawal et al., the reported pooled DY of EBUS-TBNA was 67% and 92% when EBUS-TBNA was combined with EBUS-TBFB [18]. Compared to this, the DY of EBUS-TBNA as standalone technique in our own study was lower at 61.9%. The reasons for this lower yield are unclear but may relate to the smaller mean lymph node size that was biopsied in our cohort. On average, our biopsied lymph nodes were about 0.6 mm up to 14.6 mm smaller in size than compared to the lymph nodes reported in three studies [16,22,23]. Moreover, there was a heterogeneity among the individual results for EBUS-TBNA, with two studies showing significantly lower DYs. Herth et al. showed a considerately lower overall TBNA-yield of 36% and 49% using the 22-gauge and 19-gauge needle, respectively [16]. Franke et al. reported a yield of 50% using a 22-gauge needle [22]. By adding EBUS-TBFB, the DY in our study could be improved by 23.7%. Compared to this, only Herth et al. and Franke et al. were able to show an improvement above this value, probably owing to the low DY of the EBUS-TBNA standalone technique. The overall DY of 85.7% for the combined approach in our study is reflected by similar results from previous studies, ranging from 82% up to 97% [1,16,17,22,23,24].

Regarding sarcoidosis, both techniques demonstrated a marginally lower DY for the diagnosis than reported in former studies [1,17,18]. The DY of the combined EBUS-TBNA/EBUS-TBFB technique for the diagnosis of lung cancer was comparable to results in previous studies [1,17]. However, there was no improved DY for lung cancer staging. Interestingly, in four lung cancer staging cases, EBUS-TBNA and EBUS-TBFB failed to obtain malignant tissue, while, in another two cases, only EBUS-TBNA was diagnostic.

The superior value of histological specimens compared to EBUS-TBNA cytology samples for the diagnosis of lymphoma has been reported recently [25]. Yet, the combination of EBUS-TBNA and EBUS-TBFB did not yield a significantly higher diagnostic accuracy in our study, although, in one case, a definite diagnosis of lymphoma was exclusively achieved by EBUS-TBFB. Possibly, the sample size in our study was too small to address the DY in the subgroups, particularly in lymphoma.

Another approach to provide tissue samples with greater volume in mediastinal lesions was proposed by Zhang et al. using transbronchial cryobiopsy performed under EBUS guidance [26]. They report a favorable overall DY of 91.8% for cryobiopsy in a cohort of 197 patients. Moreover, the application of a cryoprobe for transbronchial biopsy seems to be safe and shows an excellent feasibility with a successful introduction into the mediastinal lesions in 100% of their patients. Compared to this, we had to exclude 20% of our patients due to inability of penetrating the bronchial wall. Along these lines, this may offer another promising EBUS-guided diagnostic procedure to overcome the limitations of small sample volumes provided by TBNA.

Regarding our secondary endpoint, EBUS-guided TBFB permits a safe lymph node or tumor biopsy through the bronchial wall with a favorable complication rate. EBUS-TBFB was generally safe, and no major complications occurred. This finding matches previous reported complication rates [18]. Owing to the study design, minor adverse events were impossible to attribute to either TBNA or TBFB. Furthermore, in a considerable number of patients, the performance of EBUS-TBFB prevented the need for surgical biopsy. However, it should be noted that EBUS-TBFB does require bronchoscopic skills, and the performing bronchoscopists of this study were all experienced and trained in a high-throughput center specialized in interventional pulmonology.

### Limitations

In addition to the retrospective study design, there were several limitations that need to be considered. Selection bias introduced by the exclusion of patients in which EBUS-TBFB was technically not possible to perform failed to ensure that our study sample is representative of the wider patient population with unknown hilar LAD or masses. Approximately 20% of the initially enrolled patients were excluded, mostly because of the inability of the forceps to pass the bronchial wall. Given that the 19-gauge needle was used for TBNA in approximately 80% of these cases, the results might imply a correlation between needle size and technical feasibility. However, the outer diameter of both needles is identical, and this supposition will require further study. Mehta et al. used a flexible electrocautery knife to create a tract for the forceps with a successful penetration of the bronchial wall [20]. This might be a proper solution to overcome this limitation in future studies.

Compared to needle aspiration with an average sampling of three different lymph nodes per patient, only one was sampled by TBFB. Moreover, the fact that forceps biopsies were always conducted at the same location of the mass or lymph node resulted in the diagnostic accuracy of TBFB only being evaluated within one region of the lesion. Either way, TBFB may have missed diseased tissue, which we assume limits the diagnostic power of forceps biopsy.

Although the results are in favor of better quality of EBUS-TBFB specimens compared to EBUS-TBNA specimens, we did not directly assess and compare specimen quality. This study did not clarify whether any advantage of EBUS-TBFB biopsies over EBUS-TBNA specimens exists regarding the amount and quality of tissue.

## 5. Conclusions

In conclusion, our study demonstrated a safe and feasible technique for biopsy of enlarged mediastinal LAD or masses using a 1.5 mm mini-forceps in a large cohort with superior overall DY compared to EBUS-TBNA as a standalone technique. Among the subgroups, we found a significantly superior DY for the diagnosis of lung cancer and sarcoidosis. However, regarding the lower DY of EBUS-TBNA in lung cancer diagnosis in our own study compared to the current literature, the benefit of EBUS-TBFB might have been overestimated. Using an individualized patient approach based on preprocedural examination and imaging, we recommend the use of complementary EBUS-TBFB in mediastinal/hilar LAD or masses suggestive of sarcoidosis. However, technical issues are a major obstacle. A randomized study would be needed to investigate EBUS-TBFB in comparison to EBUS-TBNA and to upcoming techniques, such as transbronchial cryobiopsy.

## Figures and Tables

**Figure 1 jcm-11-04700-f001:**
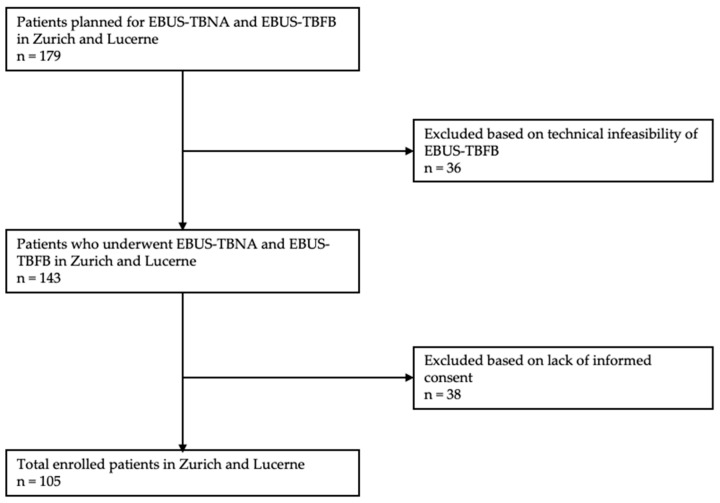
Patient flow. Abbreviations: EBUS = endobronchial ultrasound, TBNA = transbronchial needle aspiration, TBFB = transbronchial forceps biopsy.

**Figure 2 jcm-11-04700-f002:**
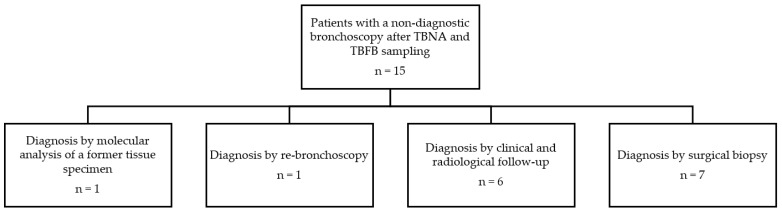
Further diagnostics or follow-up in patients with a nondiagnostic bronchoscopy after TBNA and TBFB sampling, Abbreviations: TBNA = transbronchial needle aspiration, TBFB = transbronchial forceps biopsy.

**Table 1 jcm-11-04700-t001:** Distribution and size of lymph nodes/tumors examined by EBUS-TBFB and EBUS-TBNA.

EBUS-TBFB (Lymph Node *n* = 107 Tumor *n* = 11)		19 G EBUS-TBNA (Lymph Nodes *n* = 130 Tumor *n* = 2)		22 G EBUS-TBNA (Lymph Node *n* = 152 Tumor *n* = 9)	
Lymph node station/Tumor		Lymph node station/Tumor		Lymph node station/Tumor	
4R	7	4R	30	4R	30
11R	24	11R	22	11R	26
4L	4	4L	10	4L	21
11L	21	11L	23	11L	30
7	48	7	42	7	41
10R	1	2R	2	10R	1
12L	2	12L	1	10L	2
Tumor	11	Tumor	2	12L	1
				Tumor	9
Size of Lymph node		Size of Lymph node		Size of Lymph node	
≥10–20 mm	72	≥10–20 mm	108	≥10–20 mm	70
≥21–30 mm	22	≥21–30 mm	13	≥21–30 mm	54
≥31 mm	13	≥31 mm	9	≥31 mm	28
Size of Tumor		Size of Tumor		Size of Tumor	
≥10–20 mm	2	≥10–20 mm	0	≥10–20 mm	2
≥21–30 mm	4	≥21–30 mm	0	≥21–30 mm	4
≥31 mm	5	≥31 mm	2	≥31 mm	3

Abbreviations: EBUS = endobronchial ultrasound-guided, TBNA = transbronchial needle aspiration, TBFB = transbronchial forceps biopsy.

**Table 2 jcm-11-04700-t002:** Baseline characteristics. Data are presented as the mean ± SD or *n* (%).

Characteristics (*n* = 105)	
Mean Age at Intervention	63.1 ± 13.6
Gender	
Female	37 (35.2)
Male	68 (64.8)
Smoking	
Never smoker	44 (41.9)
Active smoker	31 (29.5)
Former smoker	30 (28.6)
Concomitant diseases	
Cardiovascular disease	59 (56.2)
Respiratory disease	34 (32.4)
Renal disease	11 (10.5)
Neurological disease	19 (18.1)
Prior malignancy	29 (27.6)
Diabetes	13 (12.4)

**Table 3 jcm-11-04700-t003:** Indications for EBUS-TBFB and the final diagnosis.

	Final Diagnosis				
	Lung Cancer	Other Malignancies ^1^	Lymphoma	Sarcoidosis	Other
Total (*n* = 105)	45	7	8	27	18
Indication					
Lung cancer diagnosis ^2^ (*n* = 41)	33/41 (80.5)	3/41 (7.3)	1/41 (2.4)	1/41 (2.4)	3/41 (7.3)
Staging (*n* = 15)	11/15 (73.3)	2/15 (13.3)	1/15 (6.7)	0	1/15 (6.7)
Sarcoidosis (*n* = 36)	0	2/36 (5.6)	1/36 (2.7)	24/36 (66.7)	9/36 (25)
Lymphoma (*n* = 9)	1/9 (11.1)	0	5/9 (55.6)	2/9 (22.2)	1/9 (11.1)
Infection ^3^ (*n* = 3)	0	0	0	0	3/3 (100)
Other (*n* = 1)	0	0	0	0	1/1 (100)

^1^ Including metastasis of esophageal cancer, thymus carcinoma, melanoma, breast cancer, sarcoma, and ovarian carcinosarcoma; ^2^ including NGS (next-generation sequencing) and PD-L1 (programmed cell death ligand-1) testing; ^3^ including tuberculosis. Abbreviations: EBUS-TBFB = endobronchial ultrasound-transbronchial forceps biopsy.

**Table 4 jcm-11-04700-t004:** Diagnostic yield of TBNA and TBNA plus TBFB overall and divided into the final diagnosis.

	Total	TBNA	TBNA plus TBFB	*p*-Value
Overall	105	65/105 (61.9)	90/105 (85.7)	<0.001
Final diagnosis				
Lung cancer	34	26/34 (76.5)	33/34 (97.1)	0.016
Lung cancer staging	11	7/11 (63.6)	7/11 (63.6)	>0.05
Other malignancies *	7	3/7 (42.9)	4/7 (57.1)	>0.05
Lymphoma	8	5/8 (62.5)	6/8 (75)	>0.05
Sarcoidosis	27	12/27 (44.4)	23/27 (85.2)	0.001
Other	18	12/18 (66.7)	17/18 (94.4)	>0.05

* Including metastasis of esophageal cancer, thymus carcinoma, melanoma, breast cancer, sarcoma, and ovarian carcinosarcoma. Abbreviations: TBNA = transbronchial needle aspiration, TBFB = transbronchial forceps biopsy.

## Data Availability

The data presented in this study are openly available upon reasonable request.

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
