# Peer review of "Endobronchial Ultrasound-Guided Transbronchial Forceps Biopsy: A Retrospective Bicentric Study Using the Olympus 1.5 mm Mini-Forceps"

_jcm, 2022, doi:10.3390/jcm11164700_

Round 1
Reviewer 1 Report
I suggest to add some recommendations concerning the daily use of EBUS-TBFB according to the results of the study.
Author Response
Point 1: I suggest to add some recommendations concerning the daily use of EBUS-TBFB according to the results of the study.
Response 1: Thank you very much for the review of our manuscript.
A recommendation for the daily use of EBUS-TBFB has been added within the section “conclusions”.
Reviewer 2 Report
This is a large report evaluating the diagnostic yield and safety of EBUS-TBFB using 1.5-mm forceps. The collection of large specimens is required in clinical practice, and the significance of this report is great. However, there are some points to be improved in this report.
Major points
#1 In lines 144-146 of the text, you mentioned that EBUS-TBFB was technically impossible in 36 cases, mostly because of the inability to penetrate the bronchial wall. Is it possible that EBUS-TBFB would have been possible if the puncture opening had been larger? I think it would be desirable to describe in the text the size of the puncture needle in the cases in which the needle could not pass through the bronchial wall, and to include a discussion of this in the text.
#2 It would be preferable to show in a table the size of the puncture needle in all cases, the diameter of the target lesion (divided into 3-4 ranges), the location of the target lymph node (4R, 4L, 7, etc.), and the location of the target tumor. Please correct the table.
Minor points
#1 Lines 184-193 of the text describe the progress of cases with an undetermined diagnosis. The text alone is difficult to understand, so a diagram would make it easier to understand.
Author Response
Major Point 1: In lines 144-146 of the text, you mentioned that EBUS-TBFB was technically impossible in 36 cases, mostly because of the inability to penetrate the bronchial wall. Is it possible that EBUS-TBFB would have been possible if the puncture opening had been larger? I think it would be desirable to describe in the text the size of the puncture needle in the cases in which the needle could not pass through the bronchial wall, and to include a discussion of this in the text.
Response Major Point 1: Thank you very much for the review of our manuscript.
A description of the used puncture needles among the 36 patients without sufficient penetration has been added in lines 147-148. Furthermore, a discussion concerning these results and the proposition of a solution to overcome the above mentioned limitations is specified in lines 301-305.
Major Point 2: It would be preferable to show in a table the size of the puncture needle in all cases, the diameter of the target lesion (divided into 3-4 ranges), the location of the target lymph node (4R, 4L, 7, etc.), and the location of the target tumor. Please correct the table.
Response Major Point 2: As proposed we created 3 different tables to list the distribution and sizes of the lymph nodes/tumors examined by EBUS-TBFB and EBUS-TBNA (Table 1). Unfortunately, the location of the target tumor has not been specified in our study data, therefore we’re not able to present this information in the table.
Minor Point 1: Lines 184-193 of the text describe the progress of cases with an undetermined diagnosis. The text alone is difficult to understand, so a diagram would make it easier to understand.
Response Minor Point 1: The text has been replaced by figure 2.
Reviewer 3 Report
I think this is a good work.
I need however some clarifications:
- You said that you obteined only cytological specimens using EBUS-TBNA (specimens preserved in normal saline); why didn't you collect it as a cell-block (preserving the specimens in formalin solution for permanent fixation)? Cell-block is considered to all intents a histological preparation; probably in this way your DY of EBUS-TBNA as standalone procedure would be higher.
- Do you see differences in DY using 19 or 22 gauge needle? It could be interesting to specify this difference
- Analyzing your results (considering that in literature and in real life the DY of EBUS-TBNA for lung cancer is generally higher) I think that the real (and probably only) advantage of EBUS-TBFB is in improving the DY of sarcoidosis (and not also in lymphoma as we would have hoped): i think that you have to focus this thing in the conclusion
Author Response
Point 1: You said that you obteined only cytological specimens using EBUS-TBNA (specimens preserved in normal saline); why didn't you collect it as a cell-block (preserving the specimens in formalin solution for permanent fixation)? Cell-block is considered to all intents a histological preparation; probably in this way your DY of EBUS-TBNA as standalone procedure would be higher.
Response 1: Thank you very much for the review of our manuscript.
It is our in-house institutional protocol to preserve cytological specimens in normal saline. There has been an intern comparative trial (unpublished data) between saline and formaline resulting in no significant difference in diagnostic yield between the two techniques.
Point 2: Do you see differences in DY using 19 or 22 gauge needle? It could be interesting to specify this difference
Response 2: No significant difference was calculated. We added this information in lines 184-186.
Point 3: Analyzing your results (considering that in literature and in real life the DY of EBUS-TBNA for lung cancer is generally higher) I think that the real (and probably only) advantage of EBUS-TBFB is in improving the DY of sarcoidosis (and not also in lymphoma as we would have hoped): i think that you have to focus this thing in the conclusion
Response 3: We think that this is a fair point. A recommendation for the daily use of EBUS-TBFB according to our results has been added within the section “conclusions”.
This manuscript is a resubmission of an earlier submission. The following is a list of the peer review reports and author responses from that submission.
Round 1
Reviewer 1 Report
The manuscript describes a retrospective anaylsis of 105 patients with enlarged LN, who underwent both an EBUS-TBNA and an EBUS-guided forcpes biospy (EBUS-TBFB). The authors compared the DY of combined EBUS-TBNA and EBUS-TBFB vs. EBUS-TBNA alone. The manuscript is well written and the study was approved by the local ethic committee.
However, there are some remarks which have to be made.
The EBUS procedures has been performed by 3 physicians in 2 hospitals. For me, this is not a multicenter study like mentioned by the authors in the last paragraph of the introduction.
The authors claim, that PD-L1 testing and molecular analysis is not possible without histological specimens. That is not true, because these testings can be performed at cytological specimens as well.
A lack of the study is the high drop out rate. 38 patients didn´t sign the ICF and in 36 patients the technique was not feasible. Should be the latter included in the analysis as a procedure failure?
Furthermore the physicians took biopsies only from one representive LN whereas EBUS-TBNA was performed in multiple LN. Have been these the largest or the easiest ones? Which have been the criteria for targeting a LN for EBUS-TBFB?
The overall yield seems to be very low compared to previous EBUS-TBNA studies. However, the DY in a metanalysis comparing EBUS-TBFB and EBUS-TBNA shows only slightly higher yields. To explain the low DY of EBUS-TBNA in this trial with the LN size a comparison of the DY in subgroups based on the LN sizes should be shown.
Because the procedure is not etsablished in daily routine, images of the forceps and the procedure itself would be helpful for the reader.
At the end, the scientific impact of this study is questionable. 6 studies and one meta-analysis have been published in the recent years.
Reviewer 2 Report
General comments:
This study investigated diagnostic yield, safety and feasibility of EBUS-TBFB using the 1.5 mm mini-forceps in addition to EBUS-TBNA as compared to standard EBUS-TBNA as standalone technique for diagnosis of mediastinal/hilar LAD or masses in a multicenter study. This theme is one of the topics recently focused. However, I think this study have some problems, therefore, several modifications are necessary.
Major Comment 1:
This study is a retrospective study. However, the techniques seem to be fairly uniform despite a retrospective design and a multicenter study. Why?
Major Comment 2:
This study excluded 20% of patient due to infeasibility of EBUS-TBFB. I think it is important about what factor would be related to the feasibility (the size of target, the location of target, needle size of EBUS-TBNA, the cause of the disease, the distance from airway wall to lesion, and hole visibility). If possible, the authors should assess the points.
Major Comment 3:
As additional information, the authors should also show the diagnostic yield of only EBUS-TBFB for overall and among the subgroups.
Minor comment 1:
This study has lack of information about the location of actual targeted lymph node sampled with EBUS-TBFB, the location of actual targeted masses, and percentage of usage of each size needles.
Reviewer 3 Report
The authors showed that the use of TBFB in combination with EBUS-TBNA improved the diagnostic yield of mediastinal and hilar lesions. Forceps smaller than 1.5 mm (Boston Scientific) have been used previously, but the amount of obtained tissues was not sufficient; this report is novel in that it uses 1.5 mm sized forceps. In addition, 1.5 mm size forceps are considered to be advantageous in terms of medical cost. I consider that this article is a useful report for JCM’s readers.
However, I have a number of concerns which need to be improved. Complications are divided into minor and major bleeding - are these defined in text? Only diagnostic yields are compared in this study, but could comparisons be made for the size of tissue taken and tumour volume, as is done in many papers? In terms of cost, 1.5mm size forceps is expected to be cheaper than the mini forceps biopsy (Boston Scientific), please add that to the Discussion.